# Outcome of burn injury and its associated factors among burn patients attending public hospitals in North Showa Zone, Ethiopia: A cross-sectional study

Ejigu Mulugeta Shewaye[1], Akine Eshete Abosetugn[1], Mekasha Getnet [2], Dr. Abebe Minda[1], Abebe Nigussie Ayele[3]*, Mitiku Tefera [4]

1 Departments of Public health, Debre Berhan University, Asrat Woldeyes Health Science Campus, Debre Berhan, Ethiopia, 2 Department of Nursing, Debre Berhan University, Asrate Woldeyes Health Science Campus, Debre Berhan, Ethiopia, 3 Department of Pediatrics Nursing, Debre Berhan University, Asrate Woldeyes Health Science Campus, Debre Berhan, Ethiopia, 4 Department of Midwifery, Debre Berhan Health Science College, Debre Berhan, Ethiopia

* abebe2014nigussie@gmail.com

**Data Availability Statement:** All data are available in the manuscript and supporting file.

## Abstract

Burn injury is a major contributor to morbidity and mortality in developing countries. In Ethiopia, the outcome of burn injuries and associated factors among burn patients were not clearly described. To assess the outcome of burn injuries and its associated factors among burn patients attending public hospitals in the North, showa Zone, Ethiopia. An institution-based cross-sectional study was conducted among 420 burn patients in public hospitals of the North showa, zone. Systematic random sampling was used to select study participants. Structured checklists were used to extract data from burn patients' medical records. Data was entered using Epi-Data version 4.6. Data was analyzed using SPSS version 25. A p-value of $\leq 0.05$ in the multivariable logistic regression was used to declare a significant association. In this study, the prevalence of discharges with complications was 40.9% (95% CI: 36.5–45.6). The odds of developing complications among patients having pre-hospital intervention were nearly four times the odds of not having the intervention (AOR = 3.8, 95% CI, 1.11–13.25). The odds of developing complications among patients having scalds were four times the odds of not having scalds (AOR = 4.3, 95% CI, 1.52–12.32). A patient who received fluid and electrolytes was 76% less likely to develop the outcome of burn injury discharged with burn complications. Patients with TBSA less than 20% were 66% less likely to be discharged with complications compared to patients with TBSA greater than 20%.: This study demonstrates a significantly higher level of outcome for patients with burn injuries who were discharged with complications, leading to death and other bad outcomes. Therefore, stakeholder would more emphasis in health education on prevention of burn injuries, first aid treatment of burn, treatment of the cause of burns, and providing fluid and electrolytes.

**Funding:** The authors received no specific funding for this work.

**Competing interests:** The authors declare that they have no competing interests exist.

## Introduction

Burn is a term used to describe skin and tissue damage brought on by a fire, scald, electrical shock, chemical burn, or radioactive radiation. Due to the prolonged hospitalization for rehabilitation and the treatment of wounds and scars, burns are the most expensive traumatic injuries [1]. Large burn victims have many of the same physiological reactions as victims of major trauma, particularly venous thromboembolism [2]. It is one of the most frequent and unpleasant forms of trauma, in addition to being a substantial worldwide health problem that increases morbidity and death and causes significant material, psychological, and financial damage [3]. Burn-related wounds are damaging injuries that are linked to morbidity, a reduction in quality of life, and emotional well-being [4]. Mortality and morbidity from burns remain high, particularly in developing countries [5]. These areas are also areas where the lack of socio-economic growth, and geopolitical instability create additional barriers not only to the delivery of health care but also to the acquisition of continuous professional development [6]. Burns are estimated to be the root cause of 180,000 deaths worldwide, most of which currently occurs in low- and middle-income countries [7]. Burns are the fourth-leading cause of trauma in the world and are common in developing countries [8]. Burn injury incidence and mortality affect all regions of the world, but are most concentrated in LMICs [9]. Burns are a major public health problem worldwide [10]. The fatality rate for child burn victims in Africa was 27.8% [11].

A study shows that the magnitude of burns with complications in Ethiopia was 14.9%. The highest risk groups were children and older age groups compared to adults [12]. In almost all age groups, the prevalence of burns is higher in women [13]. The increase in burn risk in 3rd degree burns was more likely in cases of wound infection, and the risk of such infection was also considered in burn surgery [14]. Wound infection and septicemia were the two most common complications in burn patients [15]. A burn injury contains both nociceptive and neuropathic pain components, so managing a person who suffers from a burn injury is one of the most difficult challenges for first responders [16]. Treating a burn requires many hours of wound care by nursing staff, possibly multiple surgical procedures, and expensive hospitalization [17]. Burns are the most painful, require special attention, and are considered a major public health problem [18]. Burns are the fourth most common type of trauma causing death and disability after traffic accidents, falls, and interpersonal violence [19]. Globally, burns are one of the leading causes of disability, accounting for more than 8 million disability-adjusted life years (DALYs) [20]. In South Africa, an estimate of US$26 million is spent annually on burn injuries, but indirect costs such as lost wages, long-term treatment for disfigurement and emotional trauma, and the contribution of family resources also indirectly contribute to socio-economic impacts [21]. Currently, most burn centers around the world care for patients based on a protocol of burn management, like emergency phase management, acute wound care, and rehabilitation modalities [22].

In the 20th century, advances in burn care led to an increased focus on reducing burn morbidity [23]. Burns often result in significant psychological, educational, and social stigma, and the resulting changes are exacerbated by many factors, including the circumstances of the burn, the severity and location of the injury, the personality characteristics of the injured person, access to supportive interpersonal communication, and social relations [24]. Both the patients and their families may experience physical and emotional suffering as a result of the burns and treatment [25]. Burns are the second most common injury in rural Nepal, accounting for 5% of hospital admissions [7]. It is estimated that over a million people in Africa each year, and 18% of hospitalizations are due to burns [26]. Burns are a common source of harm in developing countries, including Ethiopia [27]. The incidence of burns varies by country and

population group, and the severity of the burn depends on the degree of heat, duration of exposure, and thickness of the skin [28]. Burns occur mainly at home and in the workplace, as in Bangladesh and Ethiopia, which show that 80–90% of burns occur at home [29]. In Ethiopia, the outcome of burn injuries and associated factors among burn patients were not described. Hence, it needs more study to assess the possible outcome of burns and support a uniform treatment protocol in health facilities. Therefore, this study assessed the outcome of burn injuries and their associated factors in a public hospital in North Shewa Zone, Amhara regional state, Ethiopia.

## Methods

### Ethical statement

Ethical approval and clearance were obtained from the institutional review board (IRB) of Asrat Woldeyes Health Science Campus. Permission was obtained from the Zonal Health Office and the selected hospital chief executive officer. All aspects of basic ethical research principles were addressed, so the study participants were selected based on the research requirements. The study participant's name was not written in the questionnaire's form and will never be used in connection with any information you tell us. All information given by the study participant was kept strictly confidential. Codes and aggregate reporting were used to eliminate names and other personal identifiers of respondents throughout the study process to ensure anonymity.

### Study design, setting and period

An institution-based cross-sectional study was conducted among 420 burn patients in a public hospital in the North Showa Zone, Northeast Ethiopia, from April to May in, 2023. Debre Berhan City is located 130 kilometers from Addis Ababa, the capital city of Ethiopia, and 696 kilometers from Bahirdar, north east, the regional state of Amhara. The total population of North Showa Zone was 2,322,148, of whom 1,171,510 were men and 1,150,638 were women in 2019. According to the Zonal Health Office Report, North Shewa has 164 private clinics, 97 governmental health centers, 391 health posts, 8 primary hospitals, of which two are private primary hospitals, 2 general hospitals, and one comprehensive specialized hospital. North Shewa Zone has a total of 10 public and 2 private hospitals.

### Study population

All burn patients who attended burn clinics have medical record folders in selected public hospitals from January 1st, 2018 to December 30th, 2022, in the North Showa Zone, Amhara, Ethiopia.

### Inclusion and exclusion criteria

**Inclusion criteria.** All burn patients who attended North Shewa Zone Public Hospital registered on patient cards or medical record folders in the hospital, and whose card or record was available in the medical record unit or card room, were included.

### Exclusion criteria

Burn patients with incomplete records or information on either the burn or missed diagnosis upon review were excluded from the study.

## Conceptual frame work

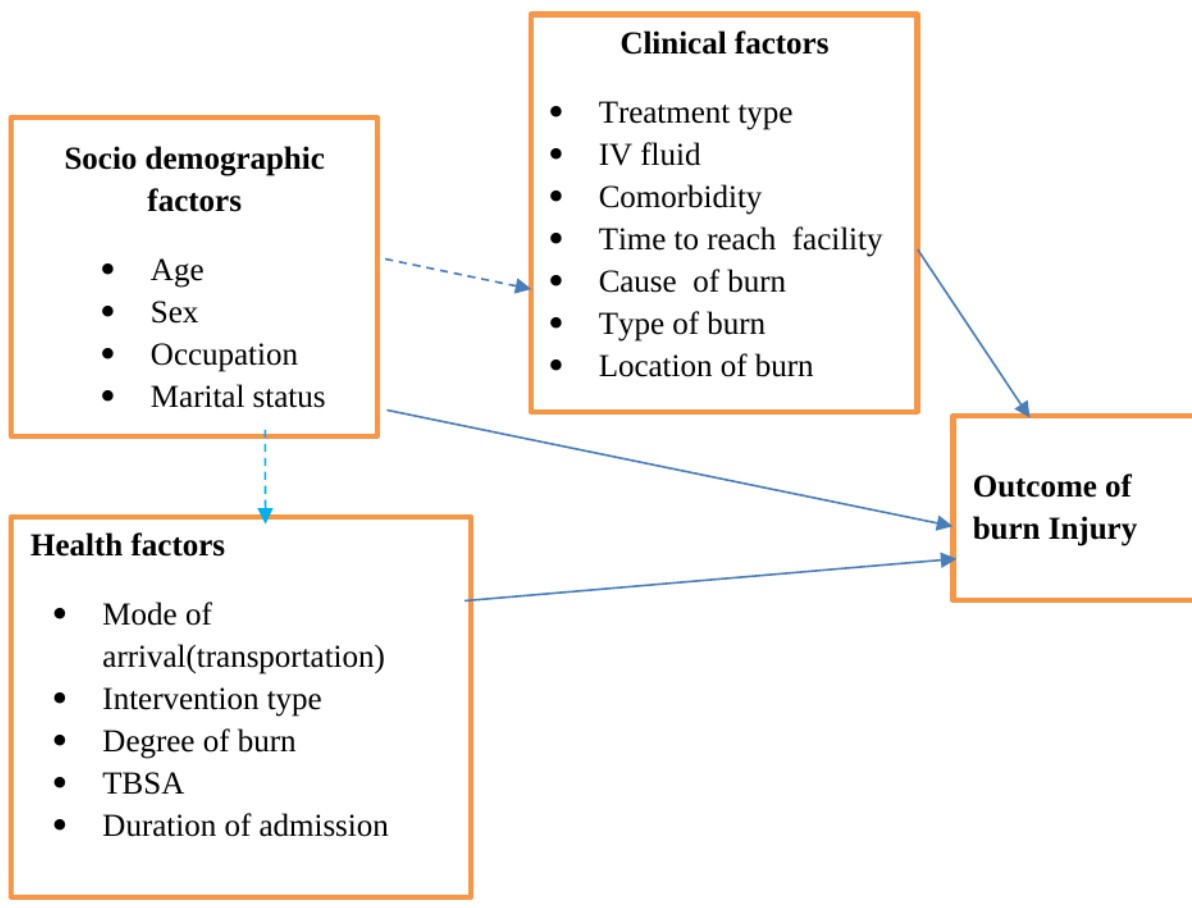

**Fig 1. Schematic presentation of sampling procedure for a study on the outcome of burn injury and its associated factors among burn patient at public hospital in North Shewa Zone, 2023.**

### Sample size determination and sampling techniques

The sample size for this study was determined by single-population formulas. Sample size was calculated using the prevalence of burn injury (p = 54.7%) [12], with a 5% margin of error, a 95% CI, and a 10% non-response rate were considered. By adding a 10% non-response rate, the total sample size becomes 420.

There are 10 public hospitals in the North Showa Zone; from these, five were selected using a simple random sampling method. The medical record numbers (MRN) of the burn patient were used as a sampling frame. The sample was allocated proportionally to each hospital. Study participants were selected using a systematic random sampling technique. First, determine the sampling interval (K) value by dividing the total admitted burn patients in the study period by the total sample size, which gives 2.17 ≈ 2, as shown in (Fig 1).

## *Sampling techniques*

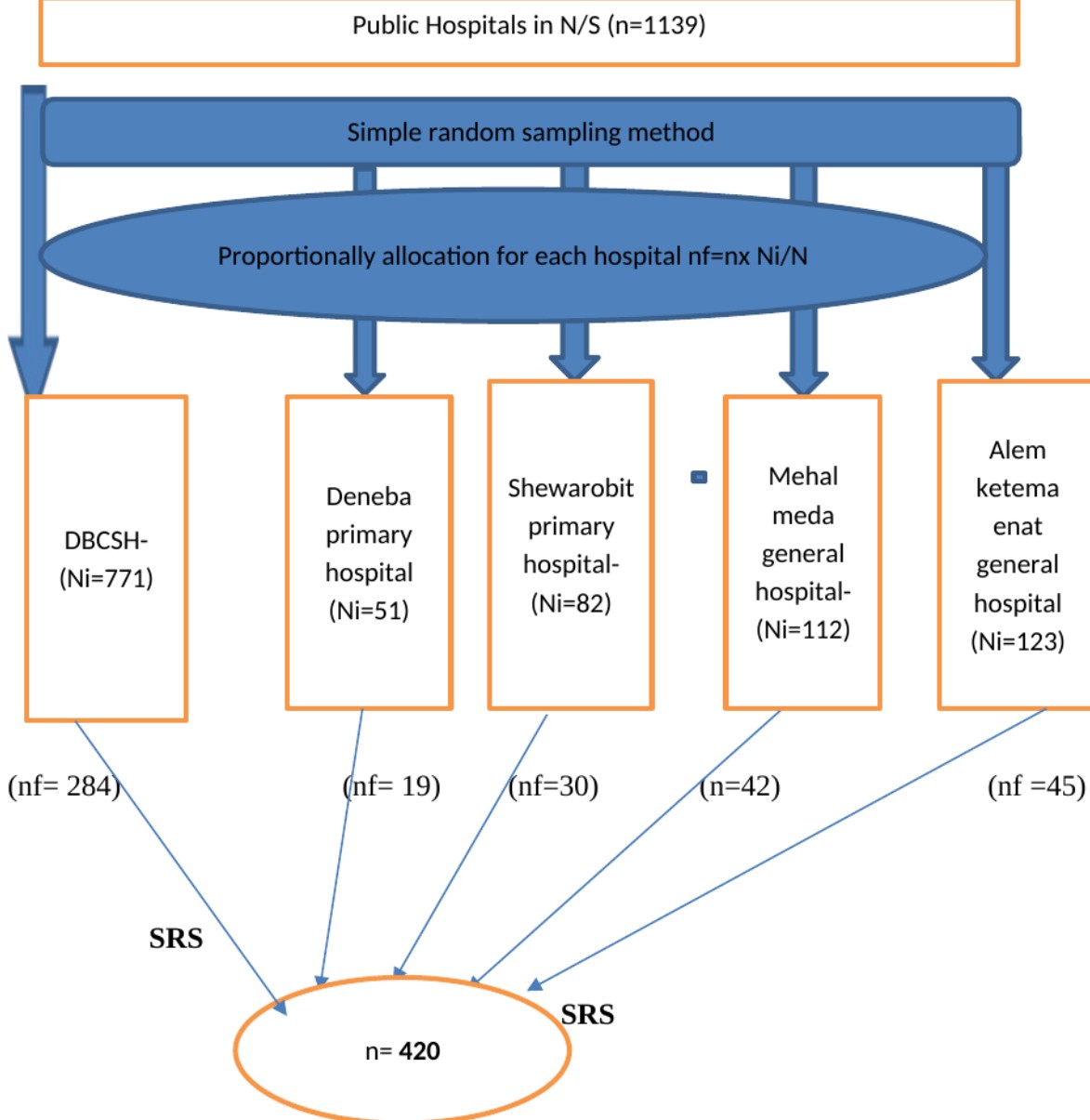

**Fig 2. Conceptual frame work for a study on the outcome of burn injury and its associated factors among burn patient at public hospital in North Shewa Zone, 2023.**

### Variables

Our dependent is outcome of burn injury discharge with complication or without complication. The independent variables are socio-demographic factors, health factors and clinical factors. These variables are described in conceptual frame work as shown in (Fig 2).

## Operational definitions

**Discharge with complications.** In this study, the patients have been discharged with one of the following: contracture, disfigurement, amputation, scar of skin graft, or death, as reported on the chart, which is considered a poor outcome of burn injury.

## Burn patients

A patient who sustained a burn injury.

## Burn injury

An injury to the skin or other tissue caused by thermal, radiation, chemical, or electrical [30].

**Type of burn injury.** Classified by depth of skin layers as superficial, deep partial thickness, full thickness, and total body surface area burned is involved.

## Outcome of burn injury

After the patients admitted to the hospital are diagnosed as discharged with or without complications, as described by physician that reported in chart [31].

## Data collection procedures

Data were collected using a structured checklist adopted from national burn patients' medical record formats (patient cards) available in the medical record unit of the public hospital in the North Shewa Zone. All charts of burn patients diagnosed with burn injuries between January 1st, 2018 and December 30th, 2022, at a selective public hospital in the North Shewa Zone were reviewed from patient registries. The checklists were well prepared in the English language, a language in which burn patients' medical records or patient cards were written. It contains information on socio-demographic conditions, clinical factors, and health-related factors. Data were extracted from the patient's medical record via record review.

## Data quality control and assurance management

A pre-test was done on 5% of the study samples at Hakim Gizaw Hospital, which is assumed to have similar characteristics to the study area. After the pre-test, necessary corrections were made to the checklist to amend language translation errors and the coherence of the checklist. Close supervision was conducted by the principal investigator and one MPH in epidemiology expert who had experience in the study setting. Data collectors and supervisors were trained on ethics, data collection tools, and how to review medical records and handle the patient card itself. Data collection is well supervised on a daily basis. Filled data were checked daily by supervisors and the principal investigator to minimize the errors that were created during the review as early as possible. Data collectors reviewed recorded cards in a separate place to keep privacy and confidentiality information.

## Data analysis procedures

The collected data were checked manually for incompleteness and inconsistency. Data were entered using Epi-Data version 4.6 and analyzed using SPSS version 26. Descriptive statistics (frequency and percentage) for categorical predictors were done. The statistical association between each independent variable and the dependent variable was assessed using bivariable logistic regression. In the bivariable analysis, a predictor with a P-value<0.25 was considered for the multivariable analysis. The goodness of fitness was assessed by a Hosmer-Lemeshow

test that was not significant (0.212). The colinearity among independent variables was assessed by the variance inflation factor (1–2.5); those variables with a VIF of less than 10 were considered to have no multi-colinearity. A multivariable logistic regression analysis was conducted to identify factors significantly associated with the outcome of burn injuries. The significance of the association was declared at a p-value of $\leq 0.05$ within the 95% CI.

## Result

### Socio-demographic characteristics

A total of 408 participants participated, with a response rate of 97%. Among 408 participants, 44.9% were male. From the total of study participants, 68.3% were single, and 30.1% were married. Most of them were government employees and students. Most of the participants were at home when the burn happened, as seen in "(Table 1)."

### Health related factors

In this study, the main circumstances of surrounding burn participants were through accident (98.8%) and the remaining was by assault (1.8%). Most of the participants had superficial thickness burns (71.6%), 27.5% had partial thickness burns, and only 1% had full thickness burn injuries. Some participants (5.1%) had comorbidities with epilepsy. Among the study participants, 11.3% had pre-hospital interventions described as "(Table 2)".

### Clinical characteristics of participants

Of the participants admitted to the hospital, 75.2% were staying less than fifteen days. During hospital stay, 22.1% participants took fluid and electrolyte. 22.3% had wound management, and most of them used antibiotics. From the participant's outcomes of burn injury with complications, 15.1% had disability, 20.5% had died, and 64.4% had been scarred, as shown in "(Table 3)".

**Table 1. Socio-demography characteristics of participants at public Hospitals in North Shewa Zone, Amhara Region, Ethiopia, in 2023 (n = 408).**

| Variables | Category | Frequency | Percent |
|---|---|---|---|
| Sex | Male | 183 | 44.9 |
| | Female | 225 | 55.1 |
| Age of participant | ≤15 year | 208 | 51.0 |
| | 16–40 year | 174 | 42.6 |
| | ≥41 year | 26 | 6.4 |
| Marital status(n = 252) | Single | 172 | 68.3 |
| | Married | 75 | 30.1 |
| | Widowed | 5 | 1.6 |
| Occupation(n = 254) | Government employed | 54 | 21.5 |
| | Non-governmental organization | 44 | 17.5 |
| | Merchant | 27 | 10.8 |
| | Farmer | 28 | 11.6 |
| | Daily laborer | 5 | 1.6 |
| | House wife | 41 | 16.3 |
| | Student | 52 | 20.7 |
| Where a patient was found when burn occurred | Home | 365 | 89.5 |
| | Street | 8 | 2 |
| | School | 28 | 6.9 |
| | Work place | 7 | 1.7 |

**Table 2. Health-related factors of participants at public hospitals in the North Shewa Zone, Amhara Region, Ethiopia, in 2023 (n = 408).**

| Variables | Category | Frequency | Percent |
|---|---|---|---|
| The main circumstance of surrounding burn | Accident | 403 | 98.8 |
| | Assault | 5 | 1.8 |
| How was the depth of burn injury | Superficial | 292 | 71.6 |
| | Partial thickness | 107 | 27.5 |
| | Full thickness | 9 | 1 |
| How much body part affected via burn | ≤20% | 319 | 79 |
| | ≥21% | 89 | 21 |
| Were your neck and head affected | Yes | 168 | 57.6 |
| | No | 240 | 42.4 |
| Were your upper extremity affected | Yes | 161 | 39.4 |
| | No | 247 | 60.6 |
| Were your lower extremity affected | Yes | 64 | 15.7 |
| | No | 344 | 84.3 |
| Were your anterior trunk affected | Yes | 9 | 2.2 |
| | No | 399 | 97.8 |
| Were you have comorbidity (Epilepsy) | Yes | 21 | 5.1 |
| | No | 387 | 94.9 |
| Were you have pre- hospital intervention | Yes | 42 | 11.3 |
| | No | 362 | 88.7 |

## Cause of burn

This study showed that flame (35.5%) was the most common cause of burn injury, hot liquids (27.8%) were the second cause of burn, steam (13.2%) was the third, chemical (10.7%) burn was the fourth, and electrical (12.8%) was the last cause of burn, as seen in"(Fig 3)".

## Outcome of burn injury

In the study, the proportion of the outcome of burn injury discharged with complications was 40.9% (95% CI: 36.5–45.6), as seen in "(Fig 4)". The main adverse complications of burn injury were 12 (8%) deaths, 22 (15.3%) disabilities, and 112 (76.7%) contractures.

**Table 3. Clinical factors of participants at public hospital in the North Shewa Zone, Amhara Region, Ethiopia, in 2023 (n = 408).**

| Variables | Category | Frequency | Percent |
|---|---|---|---|
| Used fluid and electrolyte | Yes | 90 | 22.1 |
| | No | 318 | 77.9 |
| Used burn wound management | Yes | 91 | 22.3 |
| | No | 317 | 77.7 |
| Used antibiotic | Yes | 370 | 90.7 |
| | No | 38 | 9.3 |
| Used pain management | Yes | 326 | 79.9 |
| | No | 82 | 20.1 |
| Length of stay in hospital (n = 303) | ≤ 15day | 228 | 75.2 |
| | 16–25 day | 25 | 8.3 |
| | ≥26 day | 50 | 16.5 |
| If burn outcome with complication (n = 146) | Death | 30 | 20.5 |
| | Disability | 22 | 15.1 |
| | Scaring | 94 | 64.4 |

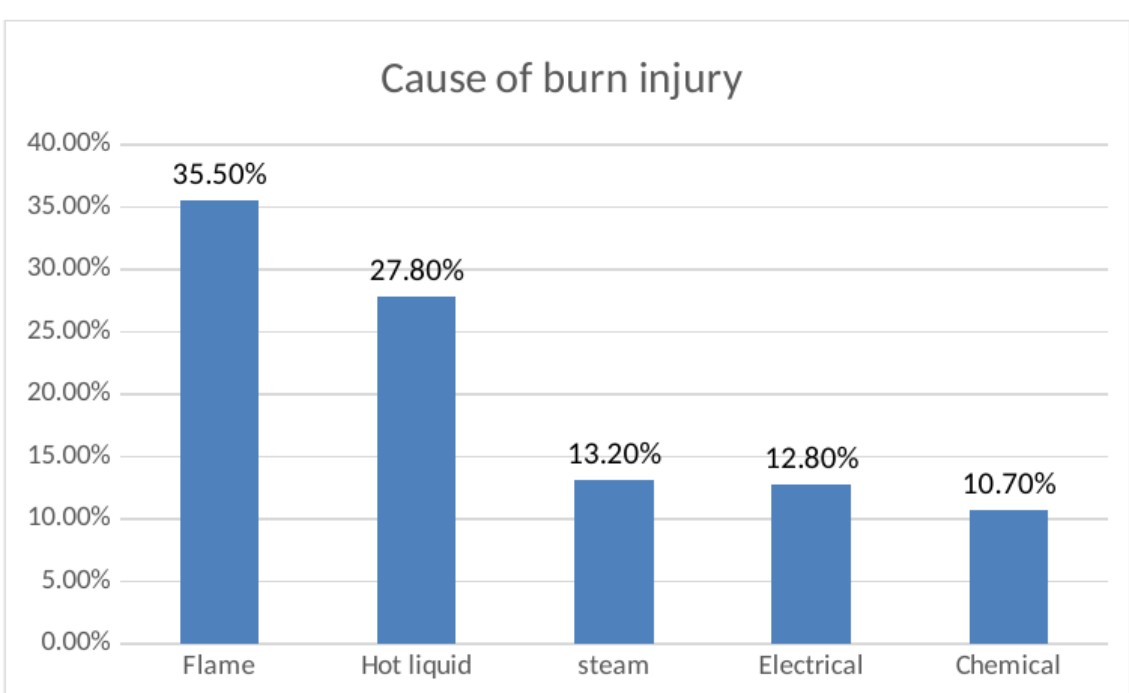

**Fig 3. Schematic presentation of cause of burn among burn patients attending at public Hospitals in North Shewa Zone Amhara Region Ethiopia 2023 (n = 408).**

### Multivariate logistic regression

The odds of developing complications among patients having pre-hospital intervention were nearly four times the odds of did not have the intervention (AOR = 3.8, 95% CI, 1.11–13.25). Similarity, the study reveals that the odds of developing complications among patients with scalds were four times the odds of did not have scalds (AOR = 4.3, 95% CI, 1.52–12.32).

Patients who had provided fluid and electrolyte were 76% less likely to experience the outcome of burn injury discharged with complications than patients who had not used fluid and electrolyte (AOR = 0.24, 95%CI, 0.10–0.56). Similarly, the occurrence of total body surface area (TBS) less than twenty percent burned was 66% less likely to affect the outcome of burn injury discharged with complications compared to patient's whose total body surface area burned greater than twenty-one percent (AOR = 0.34, 95%CI, 0.14–0.82), as shown in Table 4.

### Discussion

In this study, 40.9% (95% CI: 36.5–45.6) had the outcome of a burn injury discharged with complications. Having pre-hospital intervention, patients with scald burn, early fluid and electrolyte provided, and having a total body surface area <20% were significant factors associated with discharge with complications. The result of this finding was higher as compared to studies in Ayder referral hospital, Mekelle city, Tigria regional state, and Addis Abeba Yekatit 12 hospital, Ethiopia, where 13.3% and 17.3% had poor outcomes for burn injuries [13, 31]. The possible reason might be due to the difference study area, study period variation, and small sample size. In another similar study in Addis Ababa burn, emergency, and trauma hospitals, only 14.2% were discharged with complications [12]. The difference might be that in the burn, emergency, and trauma hospitals, there was more sophisticated equipment in the emergency intensive care unit, enough emergency care health professionals, and well-equipped

## Outcome of burn injury

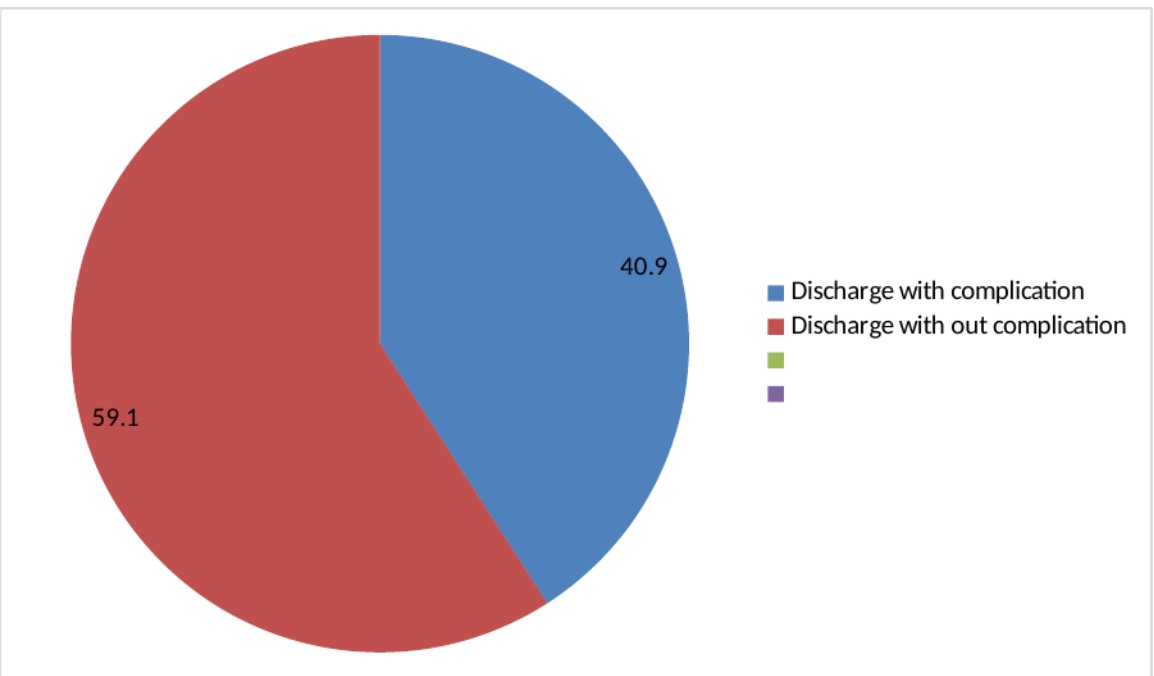

**Fig 4. Schematic presentation of the proportion on poor outcome among burn patients attending at public Hospitals in north Shewa Zone Amhara Region Ethiopia 2023 (n = 408).**

infrastructure at health institutions specifically for burn injury patients. This might be a reason for the low proportion of burned patients with the outcome of burn injury discharged with complications. But this finding is consistent with a study done at Haromia University [32]. Other studies conducted in Tanzania and Nepal revealed different results [33, 34].

The reason might be due to study population, area of study, study period, and time of data collection. In this study, the most frequent cause of injury is flame, which accounts for 35.5%, followed by scalds, steam, electrical, and chemical burns. The result of this study is consistent with studies done in Gonder, Ethiopia, and Tanzania [33, 35]. But this study contradicts studies in Addis Ababa Burn, Emergency, and Trauma Hospital, Ayder Referral Hospital, and Mekelle City [12, 31]. The possible justification might be due to population distribution by study population and socio-economic variation between populations. In this study, flame burn is the number one cause for the study population, followed by scalds, steams, electrical, and chemical burns.

In this finding, the affected body surface area (TBSA) burned due to the burn injury is an important feature in the determination of the burn injury. From the study participants, 79% of patients had a burn extent of less than 21% of TBSA. Findings of this study revealed that the occurrence of TBAS areas less than twenty percent burned were 66% less likely to discharge with compilation as compared to patients whose TBAS areas burned more than twenty-one percent. Which is in line with the study shown in Bahir Dar [13]. This results in a decline in intravascular hypervolemia, which can produce complications like ischemia in body organs.

**Table 4. Bi-variable and multivariable logistic regression on outcome of burn injury among burn patients attending public hospital in North Shewa Zone, Amhara Region, Ethiopia, in 2023(n = 408).**

| Variables | | | Outcome of burn injury | | |
|---|---|---|---|---|---|
| | | | COR (95% CI) | AOR(95% CI) | P-value |
| Age of participant | ≤15 year | 128 80 | 1.86(1.82–4.23) | 2.57(2.52–12.59) | 0.24 |
| | 16–40 year | 101 73 | 1.6(1.70–3.69) | 3.52(4.75–16.35) | 0.10 |
| | ≥41 year | 12 14 | 1 | | |
| Length of hospital stay | ≤15 day | 117 111 | 0.59(2.31–4.11) | 0.72(1.35–3.49) | 0.38 |
| | 16–25 day | 20 5 | 2.25(1.72–7.01) | 2.28(1.64–8.14) | 0.20 |
| | ≥ 26 day | 32 18 | 1 | | |
| What was the main cause of burn injury | Flame | 92 51 | 1.80(1.94–3.42) | 1.13(1.55–3.46) | 0.49 |
| | Scald | 73 38 | 1.94(1.99–3.80) | 4.33(1.52–12.32) | **0.006*** |
| | Steam | 25 29 | 0.82(1.38–2.78) | 1.80(1.50–6.49) | 0.36 |
| | Chemical | 21 22 | 0.95(1.42–2.14) | 0.62(1.18–2.08) | 0.44 |
| | Electrical | 26 26 | 1 | | |
| TBS in % | ≤20% | 42 40 | 0.62(1.38–2.06) | 0.34(1.14–2.82) | **0.017*** |
| | ≥21 | 199 118 | 1 | | |
| Sex of participants | Male | 114 69 | 1.2(1.85–1.89) | 1.17(1.65–2.09) | 0.58 |
| | Female | 127 98 | 1 | | |
| Did you used antibiotic | Yes | 214 156 | 0.55(1.26–3.16) | 0.56(1.16–3.90) | 0.35 |
| | No | 27 11 | 1 | | |
| Did you used fluid and electrolyte | Yes | 37 53 | 0.39(1.24–2.62) | 0.24(1.10–3.56) | **0.001*** |
| | No | 204 114 | 1 | | |
| Did you have pre-hospital intervention | Yes | 19 22 | 0.57(2.30–3.09) | 3.84(2.11–7.25) | **0.033*** |
| | No | 218 145 | 1 | | |
| Was your lower extremity affected | Yes | 42 23 | 1.32(1.76–2.29) | 3.38(1.96–9.88) | 0.29 |
| | No | 199 144 | 1 | | |
| Was your upper extremity affected | Yes | 128 107 | 0.66(2.44–5.99) | 0.98(3.48–6.00) | 0.96 |
| | No | 113 60 | 1 | | |
| Did you have co-morbidity(Epilepsy) | Yes | 15 6 | 1.78(1.67–4.68) | 0.71(1.17–2.98) | 0.64 |
| | No | 226 161 | 1 | | |

COR = Crude odd ratio, AOR = Adjusted odd ratio, CI = confidence interval

Similarly, patients who had received fluid and electrolyte were 76% less likely to suffer a burn injury discharged with complications than patients who had not used fluid and electrolyte during their hospital stay. The finding is also consistent with the study done in Ayder Referral Hospital, Mekelle City [31]. This could implicate that increasing public awareness regarding health-seeking behavior and early admission for intravenous infusion can diminish complications. This suggests that patients established timely supportive interventions related to fluid and electrolyte, resulting in fewer burn complications The odds of developing complications among patients having pre-hospital intervention were nearly four times the odds of did not have the intervention. This finding was not consistent with studies conducted in Haromia and Tanzania [32, 33]. The reason might be due to the difference in preexisting medical conditions before arriving at a health institution, the extent and depth of burn injuries, cultural traditions among burn management, and the quality of care in the community. The odds of developing complications among patients with scalds were more than four times the odds of did not have scalds. This finding is supported by a study conducted in South Gondar Zone government hospitals in Ethiopia [35]. But this result is different from a study done at Buganda Medical Centre

in Northwestern Tanzania that showed flame, electrical, and chemical burns were more likely to cause complications as compared to scald burns [33]. The difference might be due to large sample size, life style, and socio-economic status because of those factors, which are prone to different burn injuries.

## Recommendation

The health care workers place mainly emphasis on encouraging health education on pre-hospital intervention to prevent and perform initial intervention for patients with burn injuries. Increase the awareness and impact of scald burns for clients as well as the community. And also focus on providing fluid and electrolytes for patients who need hospital admission, and their total body surface area is affected by more than twenty percent. The researcher might do further case control studies as well as longitudinal studies that determine the association between risk factors and the outcome of burn injuries as discharge with complications.

### Limitation of the study

This study has a number of restrictions. The first drawback is that a few variables are absent from the typical burn registration books. We are unable to determine the precise number of burn patients who were hospitalized or discharged because the data is retrospective.

## Conclusion

This study demonstrates a significantly higher level of outcome for burn injuries discharged with complications, leading to increased death and other bad outcomes. Having pre-hospital intervention and the cause of burns with scalds were significant factors associated with being discharged with complications. Early fluid and electrolyte provision and having a TBS area of less than twenty percent reduce the odds for patients with a poor outcome of burn injury.

## Supporting information

**S1 Text. Annex.**
(DOCX)

**S1 Data.**
(SAV)

## Acknowledgments

We would like to thank the study participants, data collectors, and supervisors who were involved in this study and spent their valuable time responding to my study.

## Author Contributions

**Conceptualization:** Ejigu Mulugeta Shewaye.

**Data curation:** Ejigu Mulugeta Shewaye.

**Formal analysis:** Ejigu Mulugeta Shewaye.

**Funding acquisition:** Akine Eshete Abosetugn, Mekasha Getnet, Dr. Abebe Minda, Abebe Nigussie Ayele, Mitiku Tefera.

**Investigation:** Ejigu Mulugeta Shewaye, Akine Eshete Abosetugn, Mekasha Getnet, Dr. Abebe Minda, Abebe Nigussie Ayele, Mitiku Tefera.

**Methodology:** Ejigu Mulugeta Shewaye, Akine Eshete Abosetugn, Mekasha Getnet, Dr. Abebe Minda, Abebe Nigussie Ayele, Mitiku Tefera.

**Project administration:** Ejigu Mulugeta Shewaye, Akine Eshete Abosetugn, Mekasha Getnet, Dr. Abebe Minda, Abebe Nigussie Ayele, Mitiku Tefera.

**Resources:** Ejigu Mulugeta Shewaye, Akine Eshete Abosetugn, Mekasha Getnet, Dr. Abebe Minda, Abebe Nigussie Ayele, Mitiku Tefera.

**Software:** Ejigu Mulugeta Shewaye, Akine Eshete Abosetugn, Mekasha Getnet, Dr. Abebe Minda, Abebe Nigussie Ayele, Mitiku Tefera.

**Supervision:** Ejigu Mulugeta Shewaye, Akine Eshete Abosetugn, Mekasha Getnet, Dr. Abebe Minda, Abebe Nigussie Ayele, Mitiku Tefera.

**Validation:** Ejigu Mulugeta Shewaye, Akine Eshete Abosetugn, Mekasha Getnet, Dr. Abebe Minda, Abebe Nigussie Ayele, Mitiku Tefera.

**Visualization:** Ejigu Mulugeta Shewaye, Akine Eshete Abosetugn, Mekasha Getnet, Dr. Abebe Minda, Abebe Nigussie Ayele, Mitiku Tefera.

**Writing – original draft:** Akine Eshete Abosetugn, Mekasha Getnet, Dr. Abebe Minda, Abebe Nigussie Ayele, Mitiku Tefera.

**Writing – review & editing:** Ejigu Mulugeta Shewaye, Akine Eshete Abosetugn, Mekasha Getnet, Dr. Abebe Minda, Abebe Nigussie Ayele, Mitiku Tefera.

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
