## [Decision Letter · Decision Letter 0]

28 Sep 2023

PGPH-D-23-01362

Outcome of Burn Injury and its Associated Factors among Burn Patients Attending at Public Hospitals in North Showa Zone, Ethiopia: A Cross-Sectional Study

Dear Dr. Ayele,

Thank you for submitting your manuscript to PLOS Global Public Health. After careful consideration, we feel that it has merit but does not fully meet PLOS Global Public Health’s publication criteria as it currently stands. Therefore, we invite you to submit a revised version of the manuscript that addresses the points raised during the review process.

This work is important and should be published. However, there are significant issues with the manuscript draft as raised by expert peer reviewers. Please apply all reviewer's suggestion and contributions, and review the grammar and language on the manuscript to enable publication as Plos Global Public Health does not copy edit. Applying Reviewer 1's input would be very helpful and followed by the use of Grammarly, the manuscript will be enhanced in its language. Mortality should be clearly distinguished from discharged with a complication, scarring at discharge is a normal part of burns healing and reviewers will question its veracity as a complication, and other co-morbidities and current burn protocols should be included.

Also see the attached manuscript with input to improve the text.

All changes are required. Thank you so much for this important and extensive work- looking forward to working with you to publish.

With gratitude.

We look forward to receiving your revised manuscript.

Kind regards,

Barnabas Tobi Alayande

Academic Editor

Journal Requirements:

1. Please provide separate figure files in .tif or .eps format only and remove any figures embedded in your manuscript file. Please also ensure all files are under our size limit of 10MB.

2. We have noticed that you have uploaded Supporting Information files, but you have not included a list of legends. Please add a full list of legends for your Supporting Information files after the references list.

Additional Editor Comments (if provided):

Reviewers' comments:

Reviewer's Responses to Questions

**Comments to the Author**

1. Does this manuscript meet PLOS Global Public Health’s publication criteria? Is the manuscript technically sound, and do the data support the conclusions? The manuscript must describe methodologically and ethically rigorous research with conclusions that are appropriately drawn based on the data presented.

Reviewer #1: Partly

Reviewer #2: Partly

2. Has the statistical analysis been performed appropriately and rigorously?

Reviewer #1: Yes

Reviewer #2: I don't know

3. Have the authors made all data underlying the findings in their manuscript fully available (please refer to the Data Availability Statement at the start of the manuscript PDF file)?

Reviewer #1: Yes

Reviewer #2: Yes

4. Is the manuscript presented in an intelligible fashion and written in standard English?

Reviewer #1: No

Reviewer #2: No

5. Review Comments to the Author

Reviewer #1: Before being accepted for publication, the manuscript needs to be extensively revised to strengthen its language. I have provided my suggested edits for areas needing attention (highlighted in yellow). The authors may consider them during revision.

Reviewer #2: I appreciate the opportunity to review this manuscript. I applaud the authors for this study and believe it is important to understand the factors affecting outcomes of burn injuries, especially in the resource limited settings. However, I have several points of feedback:

1. Regarding patients with mortality as an outcome, it’s important to exclude them from the ‘discharged with a complication’ category. More information on these cases should be provided for both groups. Additionally, clarification is needed on what was meant by a ‘scar of skin graft,’

2. Clarification required on the assessment of burn injuries: The variables used appear to be quite broad, making it challenging to distinguish between actual complications and well-healed post-burn scars. In line 239, more information on how the researchers extracted details regarding specific complications should be provided. For instance, the variable "Did your upper extremity affected?" is mentioned, but it lacks specification regarding affected joints and range of motion. This omission would hinder the ability to determine whether a post-burn contracture, or an amputation actually occurred.

Furthermore, categorizing "scarring at discharge" as a complication might be misleading, as this is a normal part of the healing process following a burn injury.

3. The study appears limited in its consideration of comorbidities, focusing only on epilepsy. It’s essential to include other comorbidities like diabetes and heart disease, as they can significantly affect outcomes.

4. In the discussion section, it would be valuable for the authors to outline the burn treatment protocols followed at the hospitals studied. These protocols can have a substantial impact on burn outcomes.

5. The conclusion (line 292) mentions “a significantly higher level of outcome for burn injuries discharged with complications, leading to death and other worst outcomes.” The phrasing here is unclear. Please rephrase it for proper reporting.

6. Lastly, there is a need for extensive grammar editing throughout the manuscript.

6. PLOS authors have the option to publish the peer review history of their article (what does this mean?). If published, this will include your full peer review and any attached files.

**Do you want your identity to be public for this peer review?** For information about this choice, including consent withdrawal, please see our Privacy Policy.

Reviewer #1: **Yes: **Dehao Chen

Reviewer #2: No

---

## [Decision Letter · Decision Letter 1]

30 Jan 2024

PGPH-D-23-01362R1

Outcome of Burn Injury and its Associated Factors among Burn Patients Attending Public Hospitals in North Showa Zone, Ethiopia: A Cross-Sectional Study

Dear Dr. Ayele,

Thank you for submitting your manuscript to PLOS Global Public Health. After careful consideration, we feel that it has merit but does not fully meet PLOS Global Public Health’s publication criteria as it currently stands. Therefore, we invite you to submit a revised version of the manuscript that addresses the points raised during the review process.

A key challenge with this submission is the difficulty it presents to the reader due to the numerous grammatical errors. The authors should kindly pass the entire submission including the responses, cover letter etc through a spell check program like Grammarly https://www.grammarly.com/ This grammarly is a free version, and will greatly help improve the quality of the submission as several things are unclear in the text. In addition, if a collaborator with a English as a higher second or first language can be identified, authors can ask such a person to review the draft prior to submission. This is a challenge for many of us researchers from the Global South. Editing the text carefully will help make the manuscript more readable and less distracting for reviewers. Currently, the text is unclear, not grammatically correct, and ambiguous. Please work on this, as the world needs to hear the good work which you are doing.

Please note that all changes required by the reviewers are essential for acceptance (note reviewer's comment, and present rebuttal in Cover)- please recheck the reviewers' initial comments so as to comply. They are serving to improve the manuscript and have added in valid critique that should be incorporated.The rebuttal should address all previous suggestions

We look forward to receiving your revised manuscript.

Kind regards,

Barnabas Tobi Alayande

Academic Editor

Journal Requirements:

Additional Editor Comments (if provided):

Reviewers' comments:

Reviewer's Responses to Questions

**Comments to the Author**

1. If the authors have adequately addressed your comments raised in a previous round of review and you feel that this manuscript is now acceptable for publication, you may indicate that here to bypass the “Comments to the Author” section, enter your conflict of interest statement in the “Confidential to Editor” section, and submit your "Accept" recommendation.

Reviewer #1: (No Response)

Reviewer #2: (No Response)

2. Does this manuscript meet PLOS Global Public Health’s publication criteria? Is the manuscript technically sound, and do the data support the conclusions? The manuscript must describe methodologically and ethically rigorous research with conclusions that are appropriately drawn based on the data presented.

Reviewer #1: No

Reviewer #2: Partly

3. Has the statistical analysis been performed appropriately and rigorously?

Reviewer #1: (No Response)

Reviewer #2: I don't know

4. Have the authors made all data underlying the findings in their manuscript fully available (please refer to the Data Availability Statement at the start of the manuscript PDF file)?

Reviewer #1: (No Response)

Reviewer #2: Yes

5. Is the manuscript presented in an intelligible fashion and written in standard English?

Reviewer #1: No

Reviewer #2: No

6. Review Comments to the Author

Reviewer #1: To evaluate this revision, a spot check was conducted for the sections before Result to assess if reviewers' comments/suggestions from the last round of review have been incorporated. While a few suggested edits/comments have been taken into account, the revision does not reflect other suggestions - the authors should explain why those comments were not taken into account. Examples of the areas without considering reviewer's suggestions were highlighted in the attachment. I highly recommend the authors consider the suggested edits that have not been adopted to improve the writing.

Further, seemingly a reviewer's comment was directly pasted to the manuscript (lines 80-82). Copyediting errors like this should be prevented.

Reviewer #2: Thank you for inviting me to review this manuscript. I commend the authors for the extensive work done to develop this interesting research.

However, I recommend that this manuscript undergo extensive English revision, preferably by an expert in scientific writing, to ensure better presentation before publication.

7. PLOS authors have the option to publish the peer review history of their article (what does this mean?). If published, this will include your full peer review and any attached files.

**Do you want your identity to be public for this peer review?** For information about this choice, including consent withdrawal, please see our Privacy Policy.

Reviewer #1: No

Reviewer #2: No

---

## [Editor Report · Decision Letter 2]

22 Feb 2024

PGPH-D-23-01362R2

Outcome of Burn Injury and its Associated Factors among Burn Patients Attending Public Hospitals in North Showa Zone, Ethiopia: A Cross-Sectional Study

Dear Dr. Ayele,

Thank you for submitting your manuscript to PLOS Global Public Health. After careful consideration, we feel that it has merit but does not fully meet PLOS Global Public Health’s publication criteria as it currently stands. Therefore, we invite you to submit a revised version of the manuscript that addresses the points raised during the review process.

Thank you for taking the time to re-edit this work. The text has substantially somewhat, but the editors still have a significant concern about the language, grammar, syntax and understandability of the text of the cover letter, responses to the reviewers, and the manuscript. I have been guided by senior editors to send back the work and refer you to a couple of potential services (not PLOS, as unfortunately we do not have one).

This paper has substance, but the key challenge is language of expression. This is understandable as the authors are not primary English speakers, but it makes for hard reading, and review (the previous reviewers are not impressed at all with the language) and this does not meet the standards for Plos Global Public Health. Previous reviewers and editors have raised this language concern over the past iterations.

Examples of their responses to the reviewers include the following, as an example:

"We are appreciating for your view and thought fully review of our manuscript. Now we are intensively and deeply amend for the reviewers comment. As much possible we are kindly correct the grammatical error and address all comments."

"We are accepted and amend it."

"We are appreciating for your view and thought fully review of our manuscript. But our outcome variable is categorical and binary; we use binary logistic regression analysis methode. And also before analysis, we checked Goodness of fitness was assessed by Hosmer-Lemeshow test & interpret in rigorously"

I think authors are doing the best you can on their own and will need some help. I have recommended Grammarly, and getting an L1 English speaker to work with the authors previously, and doubt whether this has been applied to the entire manuscript and the response. 

Suggestions from the wider editorial board

1. Here is one service aimed at LMIC authors: https://sites.google.com/umich.edu/prepss

The authors should kindly use this and submit the manuscript after editing.

2. I can also refer you to a L1, English Speaking Intern from the US who is currently working in Rwanda in Global Surgery who can help contribute to English language corrections and help you pro bono. You could email him and send an editable version of the manuscript. I have secured his commitment for 5 -6 hours during which he could work with you in the grammatical corrections before your resubmission.

We believe you should not be excluded or rejected based on language, as the science is appreciated and important.

We look forward to receiving your revised manuscript.

Kind regards,

Barnabas Tobi Alayande

Academic Editor
---

## [Decision Letter · Decision Letter 3]

30 Apr 2024

PGPH-D-23-01362R3

Outcome of Burn Injury and its Associated Factors among Burn Patients Attending Public Hospitals in North Showa Zone, Ethiopia: A Cross-Sectional Study

Dear Dr. Ayele,

Thank you for submitting your manuscript to PLOS Global Public Health. After careful consideration, we feel that it has merit but does not fully meet PLOS Global Public Health’s publication criteria as it currently stands. Therefore, we invite you to submit a revised version of the manuscript that addresses the points raised during the review process.

The reviewers still agree that there is need for another round of copy-editing. this is the key thing standing between the manuscript and acceptance/production. Please pay attention to the cover letter and any supporting documents (previously these had not re-reviewed for language errors by the authors).

On reviewer working with you has directly requested that you add in track changes for ease of review, and I would encourage you to oblige so as to get the best out off this review process, all suggested changes are needed prior to acceptance.

We look forward to receiving your revised manuscript.

Kind regards,

Barnabas Tobi Alayande

Academic Editor

Journal Requirements:

Additional Editor Comments (if provided):

Reviewers' comments:

Reviewer's Responses to Questions

**Comments to the Author**

1. If the authors have adequately addressed your comments raised in a previous round of review and you feel that this manuscript is now acceptable for publication, you may indicate that here to bypass the “Comments to the Author” section, enter your conflict of interest statement in the “Confidential to Editor” section, and submit your "Accept" recommendation.

Reviewer #1: (No Response)

Reviewer #2: (No Response)

2. Does this manuscript meet PLOS Global Public Health’s publication criteria? Is the manuscript technically sound, and do the data support the conclusions? The manuscript must describe methodologically and ethically rigorous research with conclusions that are appropriately drawn based on the data presented.

Reviewer #1: (No Response)

Reviewer #2: Partly

3. Has the statistical analysis been performed appropriately and rigorously?

Reviewer #1: (No Response)

Reviewer #2: I don't know

4. Have the authors made all data underlying the findings in their manuscript fully available (please refer to the Data Availability Statement at the start of the manuscript PDF file)?

Reviewer #1: (No Response)

Reviewer #2: Yes

5. Is the manuscript presented in an intelligible fashion and written in standard English?

Reviewer #1: (No Response)

Reviewer #2: No

6. Review Comments to the Author

Reviewer #1: For revision, reviewers are looking for a version of manuscript with track changes. However, the uploaded "revised manuscript with track changes" only contains highlights without explanation. Can the authors submit the manuscript with track changes for assessment? Thanks.

Reviewer #2: I still suggest that the manuscript undergoes another extensive English editing/review before publication.

As pointed out in the previous reviews, some statements are not well put, eg. “death and other worst outcomes” under conclusions in Line 46 and Line 293

Table 3 ( line 211)-Variable “Did you use burn wound management” can be rephrased better for easy interpretation/ understanding by readers.

7. PLOS authors have the option to publish the peer review history of their article (what does this mean?). If published, this will include your full peer review and any attached files.

**Do you want your identity to be public for this peer review?** For information about this choice, including consent withdrawal, please see our Privacy Policy.

Reviewer #1: No

Reviewer #2: No

---

## [Editor Report · Decision Letter 4]

21 May 2024

PGPH-D-23-01362R4

Outcome of Burn Injury and its Associated Factors among Burn Patients Attending Public Hospitals in North Showa Zone, Ethiopia: A Cross-Sectional Study

Dear Dr. Ayele,

Thank you for submitting your manuscript to PLOS Global Public Health. After careful consideration, we feel that it has merit but does not fully meet PLOS Global Public Health’s publication criteria as it currently stands. Therefore, we invite you to submit a revised version of the manuscript that addresses the points raised during the review process.

Thank you for improving the manuscript- this has moved it over 4 revisions from "Major Revisions" to "Minor Revisions". Based on repeated feedback from reviewers a lot of the challenge to the review was readability due to grammatical errors. A number of these have been addressed, but the text still contains numerous widespread grammatical errors, which makes it difficult to appropriately review.

I have discussed with an intern who has English as his first language to support you in ensuring that the text is ready (if you would like his support)- Nigel Park (nigellewis024@gmail.com). Please email him to discuss your manuscript grammatical edits. He is willing to support pro-bono. He does not work for PLoS, as the journal has no preffered editing service. I am particularly aware of the challenge for authors in LMICs where English is not the first language. This is why we have supported you thus far. Please receive this pro-bono editing service as a desire to further improve your work.

These changes will improve readability and are necessary before we can send these off to the reviewers.

For instance- *the entire abstract is not clear* due to grammatical incoherences.

Background: Burn injury is a major contributor to morbidity and mortality in Sub-Saharan Africa, including Ethiopia. require regular interventions???. There is limited evidence on outcome burn injury status??? in Ethiopia including the study area (specify?). Objective: To assess the outcome of burn injury and its associated factors (unclear)
among burn patients attending ?a public Hospital (which one?) in North Showa zone, Ethiopia.2023 (unclear) Method: An institution-based, cross-sectional study was conducted among 420 burn patients in a public hHospital in North Showa zone, from April to May, 2023. Systematic random sampling was used to select patients’ cards???. Structured checklists were used to extract data from burn patients’ medical records. Data were entered using Epi-data EpiData version 4.6 and analyzed using SPSS version 25 variables with a p- value??? unclear of ≤ 0.25 in the bivariable logistic regression were included in the multivariable logistic regression. A p- value of ≤ 0.05 within 95% confidence interval in the multivariable logistic regression was used to declare significant association between the dependent variable and the independent variables. Adjusted odds ratio was used to show the strength of association between the dependent and independent variables. Result: In this study, 40.9% (95% CI: 36.5– 45.6) of patients were discharged with complications.  Patients who had pre-hospital intervention was nearly four times (AOR= 3.8, 95% CI, 1.11-13.25) and Patients burn with scalds was four (AOR=4.3, 95% CI, 1.52-12.32) times more (break into 2 sentences please or improve the conjunction) likely to develop discharged with complication. <Unclear
Patients who had provided fluid and electrolyte were 76% and the occurrence of total body surface area burn  <Unclear etc

These issues run through out the cover letter, and throught the text, though to a lesser extent.

I have spoken with an intern with Global Surgery experience to assist in perfecting the edits *pro bono *(for free), and with no expectations for authorship. You can reach out to him (Nigel Park), *in lieu *of an editing service, on **nigellewis024@gmail.com** . He has agreed to take the time to carry out the language editing with you. He is a native English speaker, with English as his first language. This will be a very important first pass, as the reviewers have reached out that this makes it difficult for them to review. Please make sure to reach out to him, or use an independent professional English editing service. Plos Global Public Health does not have one, unfortunately.

You are almost there. The concepts in this paper are good and this manuscript should be published, and I will walk with you, hand in hand till we get there. I will send out to reviewers for another round of reviews once you have been able to do the additional language editing.

We look forward to receiving your revised manuscript.

Kind regards,

Barnabas Tobi Alayande

Academic Editor
---

## [Editor Report · Decision Letter 5]

19 Jun 2024

PGPH-D-23-01362R5

Outcome of Burn Injury and its Associated Factors among Burn Patients Attending Public Hospitals in North Showa Zone, Ethiopia: A Cross-Sectional Study

Dear Dr. Ayele,

Thank you for submitting your manuscript to PLOS Global Public Health. After careful consideration, we feel that it has merit but does not fully meet PLOS Global Public Health’s publication criteria as it currently stands. Therefore, we invite you to submit a revised version of the manuscript that addresses the points raised.

Warm regards and thank you for working with us to get this to publication. Kindly edit the **abstract** on the submission portal to match that within the manuscript. **This is the abstract that the potential reviewers will see, and it is still not readable.** What is in the manuscript is much improved, **but the abstract (submitted on the journal portal) and the cover letter have not been changed at all**. Please edit this appropriately so that I can send this out to reviewers who have expressed disfficulty comprehending the work for grammatical reasons.

Please find below unclear aspects of the abstract highlighted.

The portal abstract currently reads " Background: Burn injury is a major contributor to morbidity and mortality in Sub- Saharan Africa including Ethiopia, require regular interventions. There is limited evidence on outcome burn injury status in Ethiopia including the study area.

Objective: To assess the outcome of burn injury and its associated factors among burn patients attending public Hospital in North Showa zone, Ethiopia.2023

Method: An institution-based cross-sectional study was conducted among 420 burn patients in public Hospital in North Showa zone, from April to May, 2023. Systematic random sampling was used to select patients’ cards. Structured checklists were used to extract data from burn patients’ medical records. Data were entered using Epi-data version 4.6 and analyzed using SPSS version 25 variables with a p- value of ≤ 0.25 in the bivariable logistic regression were included in the multivariable logistic regression. A p- value of ≤ 0.05 within 95% confidence interval in the multivariable logistic regression was used to declare significant association between the dependent variable and the independent variables. Adjusted odd ratio was used to show the strength of association between the dependent and independent variables.

Result: In this study 40.9% (95% CI: 36.5– 45.6) were discharged with complication. Patients who had pre-hospital intervention was nearly four times (AOR= 3.8, 95% CI, 1.11-13.25) and Patients burn with scalds was four (AOR=4.3, 95% CI, 1.52-12.32) times more likely to develop discharged with complication. Patients who had provided fluid and electrolyte were 76% and the occurrence of total body surface area burn <20% were 66% less likely to develop the outcome burn injury discharged with complication.

Conclusion: This study demonstrates a significantly higher level of the outcome of burn injury discharged with complication, leading to death and other worst outcomes. Having pre-hospital intervention, patients with scalds burn, Early fluid and electrolyte provided and having total body surface area <20% were significant factors associated with poor outcome of burn injury. Therefore health education on pre-hospital intervention and the cause of burn and providing fluid and electrolyte for patients total body surface area affected more than twenty percent

We look forward to receiving your revised manuscript.

Kind regards,

Barnabas Tobi Alayande

Academic Editor
---

## [Decision Letter · Decision Letter 6]

12 Jul 2024

PGPH-D-23-01362R6

Outcome of Burn Injury and its Associated Factors among Burn Patients Attending Public Hospitals in North Showa Zone, Ethiopia: A Cross-Sectional Study

Dear Dr. Ayele,

Thank you for submitting your manuscript to PLOS Global Public Health. After careful consideration, we feel that it has merit but does not fully meet PLOS Global Public Health’s publication criteria as it currently stands. Therefore, we invite you to submit a revised version of the manuscript that addresses the points raised during the review process.

Thank you for your attention to the previous rounds of reviews. One reviewer is happy with the progress enough to suggest acceptance, but a few issues remain unresolved in the text and have been noted in detail by the reviewer (see attachment). Ensure to **use track changes** for the next (but hopefully final) round of review. Kindly address the specific comments (particularly the changes included in the draft by the second reviewer) and keep track of them using the track changes feature in the word document (not only highlighting) for clarity for the same reviewer.

Also review and clarify Lines 35-37: "Patients who received total body surface area were 20% less likely to develop discharge with complications than those who had not burned"

Line 40: “Therefore, stakeholder would more emphasis in health education on pre-hospital intervention…….”

Thank you.

We look forward to receiving your revised manuscript.

Kind regards,

Barnabas Tobi Alayande

Academic Editor

Journal Requirements:

Additional Editor Comments (if provided):

Reviewers' comments:

Reviewer's Responses to Questions

**Comments to the Author**

1. If the authors have adequately addressed your comments raised in a previous round of review and you feel that this manuscript is now acceptable for publication, you may indicate that here to bypass the “Comments to the Author” section, enter your conflict of interest statement in the “Confidential to Editor” section, and submit your "Accept" recommendation.

Reviewer #1: (No Response)

Reviewer #2: All comments have been addressed

2. Does this manuscript meet PLOS Global Public Health’s publication criteria? Is the manuscript technically sound, and do the data support the conclusions? The manuscript must describe methodologically and ethically rigorous research with conclusions that are appropriately drawn based on the data presented.

Reviewer #1: Partly

Reviewer #2: Yes

3. Has the statistical analysis been performed appropriately and rigorously?

Reviewer #1: (No Response)

Reviewer #2: I don't know

4. Have the authors made all data underlying the findings in their manuscript fully available (please refer to the Data Availability Statement at the start of the manuscript PDF file)?

Reviewer #1: (No Response)

Reviewer #2: Yes

5. Is the manuscript presented in an intelligible fashion and written in standard English?

Reviewer #1: (No Response)

Reviewer #2: Yes

6. Review Comments to the Author

Reviewer #1: I appreciate the authors' consideration of my comments from the prior reviews. The manuscript looks much better now. However, before it being accepted for publication. There are still things that can be improved, which are listed as comments in the upload.

I'd appreciate if the authors can use track changes for the next (but hopefully final) round of review. Also, I find it challenging to open the uploaded figures. Can they be uploaded in a more common format (e.g., tiff, jpeg). Lastly, it'd be helpful to replace the long web links of some citations with DOIs of the cited articles.

Reviewer #2: I congratulate the authors and the editorial team for their tireless efforts to improve this great work. I suggest some minor edits in the abstract which still seem unclear, before publication.

-Line 35-37: "Patients who received total body surface area were 20% less likely to develop discharge with complications than those who had not burned"

Line 40: “Therefore, stakeholder would more emphasis in health education on pre-hospital intervention…….”

Congratulations, and I recommend this work to be published

7. PLOS authors have the option to publish the peer review history of their article (what does this mean?). If published, this will include your full peer review and any attached files.

**Do you want your identity to be public for this peer review?** For information about this choice, including consent withdrawal, please see our Privacy Policy.

Reviewer #1: No

Reviewer #2: No

---

## [Decision Letter · Decision Letter 7]

6 Aug 2024

PGPH-D-23-01362R7

Outcome of Burn Injury and its Associated Factors among Burn Patients Attending Public Hospitals in North Showa Zone, Ethiopia: A Cross-Sectional Study

Dear Dr. Ayele,

Thank you for submitting your manuscript to PLOS Global Public Health. After careful consideration, we feel that it has merit but does not fully meet PLOS Global Public Health’s publication criteria as it currently stands. Therefore, we invite you to submit a revised version of the manuscript that addresses the points raised during the review process.

Thank you for working with the review and editorial teams over the past few months. Both reviewers have now come to a point in our iteration where the team of reviewers and editors believe that this manuscript is about set for publication relative to when it was originally received. Both reviewers independently have decided that the manuscript should be accepted based on al the hard work and edits that you have put in to this work.

There are a few recommendations of one of the reviewers that still need to be addressed, as Plos global Public Health does not offer review/editing services, and there is no other point at which these minor changes requested by the reviewer can be corrected.

Below are the few final edits identified as needed before publication by the reviewer:

1. Line 36-37: Please change the sentence that says, "Patients who received total body surface area were 20% less likely to develop discharge with burn complications than those who had not burned" to '**'Patients with TBSA less than 20% were 66% less likely to be discharged with complications compared to patients with TBSA greater than 20%**"

2. Line 212: Steam is still a form of a scald burn. Please sub-categorize scald into ''hot liquids" and "steam" in order to keep the results and figures unaffected.

3. Line 265:Replace 21% with 20%

Once this is received and I can check to confirm that you have applied it, and then, based on the current decision of reviewers, we will let you know next steps as it will be accepted for publishing.

Thank you for your consistency and your sticking with this over the past few months.

With gratitude, 

We look forward to receiving your revised manuscript.

Kind regards,

Barnabas Tobi Alayande

Academic Editor

Journal Requirements:

Additional Editor Comments (if provided):

Reviewers' comments:

Reviewer's Responses to Questions

**Comments to the Author**

1. If the authors have adequately addressed your comments raised in a previous round of review and you feel that this manuscript is now acceptable for publication, you may indicate that here to bypass the “Comments to the Author” section, enter your conflict of interest statement in the “Confidential to Editor” section, and submit your "Accept" recommendation.

Reviewer #1: (No Response)

Reviewer #2: (No Response)

2. Does this manuscript meet PLOS Global Public Health’s publication criteria? Is the manuscript technically sound, and do the data support the conclusions? The manuscript must describe methodologically and ethically rigorous research with conclusions that are appropriately drawn based on the data presented.

Reviewer #1: (No Response)

Reviewer #2: Yes

3. Has the statistical analysis been performed appropriately and rigorously?

Reviewer #1: (No Response)

Reviewer #2: I don't know

4. Have the authors made all data underlying the findings in their manuscript fully available (please refer to the Data Availability Statement at the start of the manuscript PDF file)?

Reviewer #1: (No Response)

Reviewer #2: Yes

5. Is the manuscript presented in an intelligible fashion and written in standard English?

Reviewer #1: (No Response)

Reviewer #2: Yes

6. Review Comments to the Author

Reviewer #1: (No Response)

Reviewer #2: Congratulations to the authors, and the editorial team on the tireless efforts to make this great work much better.

Few final edits before publication:

Line 36-37: 'Patients who received total body surface area were 20% less likely to develop discharge with burn complications than those who had not burned" From your results it seems like you could change to ''Patients with TBSA less than 20% were 66% less likely to be discharged with complications compared to patients with TBSA greater than 20%"

Line 212: Steam is still a form of a scald burn. Maybe, you can sub-categorize scald into ''hot liquids" and "steam" in order to keep the results and figures unaffected.

Line 265: I think you meant 20% and not 21%?? since your categories were; below 20% and greater or equal to 20%

7. PLOS authors have the option to publish the peer review history of their article (what does this mean?). If published, this will include your full peer review and any attached files.

**Do you want your identity to be public for this peer review?** For information about this choice, including consent withdrawal, please see our Privacy Policy.

Reviewer #1: No

Reviewer #2: No

---

## [Editor Report · Decision Letter 8]

15 Aug 2024

Outcome of Burn Injury and its Associated Factors among Burn Patients Attending Public Hospitals in North Showa Zone, Ethiopia: A Cross-Sectional Study

PGPH-D-23-01362R8

Dear Mr Ayele,

We are pleased to inform you that your manuscript 'Outcome of Burn Injury and its Associated Factors among Burn Patients Attending Public Hospitals in North Showa Zone, Ethiopia: A Cross-Sectional Study' has been provisionally accepted for publication in PLOS Global Public Health.

Best regards,

Barnabas Tobi Alayande

Academic Editor

All required changes have been performed.

Thank you for this much improved work over the past few months. In my reflections, I congratulate the authors for sticking with the process in a determined and focussed manner, the reviewers for their selfless intentionality and determination not to focus on certain perceived limitations but on scholarly content, and PLOS Global Public Health for equity-driven and thoughtful journal policies that make it possible for publications like this, with merit, to be published. Great job, and hard work, all 8 revisions later.

The English language limitations of non L1/primary speakers is too often a stumbling block to publication. This should not be, and even though this road was hard and long, we have shown what is possible even without an editing service. We are not saying this is perfect work, we are saying it meets the criteria for publication in PlosGPH and the authors should be proud of the iterative, but gratifying process. As you go through the next required changes, please reflect on the fact that your reviewers and editors have walked this path together, on the same team, to bring forward this very important piece that is of improved quality and relevant to burns on the African continent.

Congratulations all!